# Microwave-Assisted Hydrodistillation of Essential Oil from *Plectranthus amboinicus*: Evaluation of Its Antifungal Effect and Chemical Composition

**DOI:** 10.3390/life13020528

**Published:** 2023-02-15

**Authors:** Oscar Antonio-Gutiérrez, José Antonio Alvízar-Martínez, Rodolfo Solano, Alfonso Vásquez-López, Sandra Luz Hernández-Valladolid, Hermes Lustre-Sánchez, Hilda Elizabet Flores-Moctezuma, Erick de Jesús de Luna-Santillana, Luicita Lagunez-Rivera

**Affiliations:** 1Laboratorio de Extracción y Análisis de Productos Naturales Vegetales, Instituto Politécnico Nacional, Centro Interdisciplinario de Investigación para el Desarrollo Integral Regional unidad Oaxaca, Santa Cruz Xoxocotlán 71230, Mexico; 2Universidad Tecnológica del Centro de Veracruz, Veracruz 94910, Mexico; 3Centro de Desarrollo de Productos Bióticos, Instituto Politécnico Nacional, Morelos 62731, Mexico; 4Centro de Biotecnología Genómica, Instituto Politécnico Nacional, Tamaulipas 88710, Mexico

**Keywords:** *Syzygium aromaticum*, *Lippia alba*, *Rosmarinus officinalis*, *Fusarium oxysporum*, carvacrol, o-cymene, caryophyllene

## Abstract

*Fusarium* wilt, a vascular syndrome in a wide range of plants, is caused by the pathogen *Fusarium Oxysporum*. The objective of this investigation was to evaluate the antifungal effect of four essential oils (EOs) (*Plectranthus amboinicus*, *Syzygium aromaticum*, *Lippia alba*, and *Rosmarinus officinalis*), which were obtained by using microwave-assisted hydrodistillation (MAH), against *F. oxysporum*. The yield obtained from *P. amboinicus* with the use of MAH was 0.2%, which was higher than that of a conventional extraction; its extraction time was also shorter. For concentrations of 100 and 300 μL/L, *P. amboinicus* caused an inhibition rate of 27.2 and 55.7%, respectively, while *S. aromaticum* caused an inhibition rate of 23.1 and 87.3%, respectively. It was observed that increasing the concentration also increased the % inhibition rate. The extracts of *L. alba* and *R. officinalis* caused an inhibition rate of 14.5 and 14.7% at 500 µL/L, respectively, at 10 days of incubation, while at this concentration, *P. amboinicus* and *S. aromaticum* achieved 100%. The major chemical compounds of *P. amboinicus* were carvacrol (41.20%), o-cymene (11.61%), caryophyllene (11.45%), α-bergamotene (7.71%), and caryophyllene oxide (4.62%), and these monoterpene hydrocarbons were responsible for the biological activity. The essential oil of *P. amboinicus* in appropriate concentrations is a potent antifungal agent that could be used for the control of *F. oxysporum*.

## 1. Introduction

Medicinal plants have been used since prehistoric times to cure different diseases throughout the world. Recently, natural products have received considerable attention and are used in the form of drugs or dietary supplements due to new discoveries and drug developments [1]. Many of these plants are being investigated in different areas, such as their application in the treatment of chronic diseases, as well as for their broad properties where antifungal properties are relevant. *Fusarium* species are molds found in the environment and various organic substrates. Some *Fusarium* species are the causes of plant pathogens, food spoilage, and mycotoxicosis in humans and animals [2]. In general, the *Fusarium oxysporum* fungus is considered one of the main causes of diseases in ornamental plants [3]. The development of *Fusarium* wilt disease arises from the interaction between host plants and *F. oxysporum*, a parasitic pathogen [4]. On the other hand, the production of ornamental plants has had a significant increase in recent years. Their main use is decorative, and they are highly appreciated for their visual characteristics [5]. Different reasons lead to the fact that *Fusarium* wilt disease continues to plague the industry, for example, imperfections in clean stock propagation systems and irrigation systems that allow propagule spread, among others. Additionally, *Fusarium* is an important pathogen of numerous crops of agricultural and economic relevance [6].

A solution to control this problem is the use of chemicals or fungicides; however, the use of fungicides causes severe damage to the environment, and some fungicides can be carcinogenic and also have a high cost [7]. An alternative to this situation is biological control, which is defined as the suppression of a disease through the application of biological products, such as microorganisms, semichemical products, and natural substances [8]. Essential oils (EOs) are a mixture of complex aromatic substances that are mainly composed of terpenes and other compounds, such as aldehydes, phenols, and ketones. EOs are characterized by a strong odor. They are derived from the secondary metabolism of plants and are considered as potential biocontrol products [9].

The fungicidal activity of EOs of clove (*Syzygium aromaticum* (L.) Merr. & L.M. Perry), lemon grass (*Cymbopogon citratus* (DC) Stapf), mentha (*Mentha* × *piperita* L.), and eucalyptus (*Eucalyptus globulus* Labill.) against *F. oxysporum* has been evaluated [10]. This investigation found that clove oil is a potent antifungal agent that could be used as a biofungicide for the control of *F. oxysporum* f. sp. *lycopersici*. Other EOs found to have promising results were obtained from *Mentha spicata* L., *Helichrysum splendidum* (Thunb.) Less. and *Cymbopogon citratus* [11]. Many *Fusarium* species are still resistant to antifungals. Thus, further research is needed on plant-derived secondary metabolites as they have significant antifungal capabilities.

One of the main methods to obtain EOs is hydrodistillation; in recent years, in order to reduce extraction time and operating costs, among other goals, microwave-assisted extraction (MAE) has been investigated [12]. MAE is a method that uses microwave energy to accelerate the extraction of bioactive compounds from plants [13]. In this technology, electromagnetic irradiation is used to heat the solid–liquid mixture so that tissues and cells are destroyed, causing the release of bioactive compounds [14]. The extraction of EOs from different plants by using MAE has been studied in recent years. For example, EOs have been obtained from *Chenopodium ambrosioides* Bert. ex Steud. and *Ocimum basilicum* L. [15]. More recently, new approaches have been evaluated, such as an enzymolysis pretreatment–MAE method [16] and microwave-assisted hydrodistillation (MAH) and extraction in situ by vegetable oil [17]. Other interesting alternatives that have been evaluated are propolis extract and the use of antagonistic fungi [18]. The search for ecological alternatives for the control of fungi that affect ornamental plants continues to be an area of interest for research. For this reason, the objective of this study was to evaluate the fungicide activities of different doses of EOs obtained by MAH of *Plectranthus amboinicus* (Lour.) Spreng., *Syzygium aromaticum*, *Lippia alba* (Mill.) N.E. Br. ex Britton & P. Wilson, and *Rosmarinus officinalis* L., as well as to identify the chemical components of the most effective essential oil against *F. oxysporum*.

## 2. Materials and Methods

### 2.1. Chemicals

Deionized water was prepared through a Milli-Q water purification system (Millipore, Waltham, MA, USA). The chemicals applied in this investigation were ACS grade from Sigma (Sigma-Aldrich, St. Louis, MO, USA).

### 2.2. Plant Material

The fresh leaves of *P. amboinicus* and *R. officinalis*, the fresh aerial parts of *L. alba*, and the inflorescence of *S. aromaticum* were collected from specimens cultivated in the surroundings of Santa Cruz Xoxocotlan, Oaxaca, Mexico. For the extraction, the material was used in a fresh state and only the inflorescences of *S. aromaticum* were used dehydrated. Two of the authors (R.S.-G. and H.L.-S.), as well as Remedios Aguilar-Santelises, the curator of the OAX herbarium (CIIDIR Oaxaca, Instituto Politécnico Nacional), were involved in the taxonomic determination of the species. A voucher specimen was prepared for each species, which was deposited in the OAX herbarium as follows: R. Solano 4420 (*R. officinalis* L.), R. Solano 4421 (*L. alba*), R. Solano 4422 (*S. aromaticum*), and R. Solano 4423 (*P. amboinicus*).

### 2.3. EOs Extraction

A domestic microwave oven with a hole in the top (MW1235WB, Samsung, Seoul, Republic of Korea) and working at a frequency output of 2450 MHz was modified for the MAH operation. The system consisted of a 2 L ball flask installed inside the microwave oven cavity and connected to a Clevenger-type apparatus that was located outside to collect the extracted EOs. Finally, a condenser connected to the Clevenger extractor complemented the system. The MAH system used in this investigation is described in Figure 1. The plant material was placed in the ball flask with water at a ratio of 1:3 *w*/*v*. The extraction time for all species was carried out until the essential oil content was exhausted, for a total time of 30 min with work periods of 2 min and rest periods of 1 min.

### 2.4. Antifungal Activity of EOs

Fungal isolates were obtained from the root tissue samples of symptomatic coriander plants (*Coriandrum sativum* L.), which were collected from commercial cultivation plots located in San Antonino Castillo Velasco, Ocotlan de Morelos, Oaxaca, Mexico. The developed colonies were purified by using the monosporic culture technique and preserved in tubes with PDA (BD Bioxon, Mexico City, Mexico) covered with sterile mineral oil. The colonies were housed in the Laboratorio de Fitopatología of the CIIDIR Oaxaca (code FoCsOax). Molecular characterization was performed using the molecular markers for elongation factor 1α (TEF-1α, primers EF1 and EF2), β-tubulin (TUB2, primers T1 and T22) [19], and ribosomal protein subunit II (RPB2, primers 5F2 and 11AR) [20]. The forward and reverse sequences obtained from the amplification of the genes were cleaned and assembled with the SeqMan v.7 program (Lasergene R, DNASTAR, Madison, WI, USA). Each assemblage underwent BLASTn homology analysis against the GenBank genetic sequence database from the National Center for Biotechnology Information (NCBI). A disc (1 cm) was cut from a 5-day-old culture of the fungus and suspended in 10 mL of sterile distilled water. To obtain the monosporic cultures, serial dilutions were prepared from the pure cultures, and individual spores were collected and grown on a PDA medium. Five doses of essential oils (100, 200, 300, 400, and 500 µL/L) were evaluated. Each dose was perfectly mixed with 1.0 L of PDA culture medium [21]. The medium was emptied into Petri dishes with a 8.5 cm diameter, and once the medium solidified, a disk with a 0.7 cm diameter of the medium with 10-day-old mycelial growth was placed in the center of each Petri dish. The Petri dishes were incubated at 25 °C for 10 days. Benomyl fungicide was used at a dose of 0.03 g/L as a positive control. The efficacy of the EOs was evaluated by estimating the % inhibition rate of mycelia, taking the control treatment as a reference. The measurements were taken every 48 h with a vernier until the mycelium from the control treatment filled the Petri dish, using the following formula [22]:% Inhibition rate = (Mc − Mt/Mc) × 100(1)
where Mc is the mycelial growth in the control, and Mt is the mycelial growth in the treatment.

### 2.5. GC-MS and Identification of Volatile Compounds

Gas chromatography–mass spectrophotometry (GC-MS) analysis was performed using an EVOQ GC-Triple Quadrupole Bruker system (456 GC, Heidelberg, Germany) equipped with an autosampler. The samples were analyzed on a fused silica capillary column BR-1ms (Bruker Daltonics, Billerica, MA, USA) with FS 30 m × 0.25 mm I.D., film thickness of 0.25 µm, and helium as the carrier gas. The injector and detector temperatures were set at 220 and 250 °C; the injected volume was 1 µL; the oven temperature was programmed at 55 °C, with increases of 35 °C/min up to 255 °C; and the ionization energy was 70 eV. Identification of the components was based on the computer-matched NIST 98 and the Dictionary of Natural Products.

### 2.6. Statistical Analysis

All determinations were conducted in triplicate. The data were analyzed using the SPSS statistical software (version 28, IBM, Armonk, NY, USA). The effects of the treatments on *F. oxysporum* were analyzed using one- or two-way ANOVA test.

## 3. Results

The yields of the different EOs obtained by using MAH are presented in Figure 2. The respective values are 0.2% for *P. amboinicus*, 3.5% for *S. aromaticum*, 0.5% for *L. alba*, and 0.3% for *R. officinalis*.

An extraction yield similar to that obtained with conventional hydrodistillation was achieved in the case of *S. aromaticum*, *L. alba*, and *R. officinalis*, but the extraction time was significantly shorter. The % inhibition of mycelial growth of the different EOs is presented in Table 1. Benomyl fungicide was used (0.03 g/L) as a positive control. In all cases of EOs, it is observed that by increasing the concentration of the essential oil, the % inhibition rate also increases. However, the extracts of *L. alba* and *R. officinalis* cause an inhibition rate of 14.5 and 14.7%, respectively, after 10 days of incubation. In the case of the essential oil of *P. amboinicus* at concentrations of 400 and 500 µL/L, a 100% inhibition rate is achieved for both concentrations. *S. aromaticum* essential oil at concentrations of 400 and 500 µL/L achieves an inhibition rate of 93.3 and 100%, respectively.

For the concentrations of 100, 200, and 300 μL/L, the essential oil of *P. amboinicus* causes an inhibition rate between 27.2% and 55.7%, while the essential oil of *S. aromaticum* causes an inhibition rate between 23.1% and 87.3% (Figure 3).

The chemical composition of *P. amboinicus* essential oil is shown in Table 2. The 27 components shown in Table 2 consist of 100% of total GC peak area. The main chemical compounds are carvacrol (41.20%), o-cymene (11.61%), caryophyllene (11.45%), α-bergamotene (7.71%), and caryophyllene oxide (4.62%). Although the main compound is Carvacrol, the presence of other compounds, such as o-cymene and others, may exert a synergistic action, which is responsible for the superior antifungal activity of the *P. amboinicus* essential oil.

## 4. Discussion

The yield obtained from *P. amboinicus* by using MAH was higher than the yield obtained by using conventional hydrodistillation. For example, the yield of essential oil from *P. amboinicus* was found to be 0.05% when using conventional hydrodistillation [23]. Another study, in which the essential oil of *P. amboinicus* was extracted by maceration with acetone, obtained a yield of 0.1% [24], while in our case, the yield was 0.2%. Due to the high water content of the plants, microwaves are absorbed quickly and strongly, which causes an increase in temperature and the destruction of plant cell walls, leading to the release of bioactive compounds [25]. In general, the extraction time was much shorter for all cases, which represents a great advantage of MAH by reducing operating costs when compared to conventional hydrodistillation [12].

The EOs of *L. alba* and *R. officinalis* did not cause the expected inhibition of *F. oxysporum*. Previous studies have shown that for some EOs, higher concentrations are required to achieve greater % inhibition, for example, a concentration of 3000 μL/L of *L. rehmannii* was necessary to achieved 100% inhibition of different pathogens [26]. Furthermore, the different activity of essential oils can be explained by the variety in water solubility and lipophilic properties that may ultimately affect antifungal activity [27]. On the other hand, the high antifungal activity of the *P. amboinicus* and *S. aromaticum* EOs could be related to their major oil components. Some studies have shown that with *O. vulgare*, the inhibitory effect is attributable to the presence of compounds with a phenolic fraction and also to the high antifungal activity of carvacrol against phytopathogenic fungi [28]. Murthy et al. [29] previously studied *P. amboinicus* essential oil for the inhibition of fungus growth. At a concentration of 500 ppm, they achieved a complete inhibition of *Aspergillus ochraceus*. In addition, the essential oil extracted from the leaves of *P. amboinicus* have shown remarkable antibacterial activity against important pathogenic bacteria [30].

*Plectranthus cylindraceus* is another species that has shown good antifungal activity, and Marwah et al. [31] demonstrated that the growth of all phytopathogenic fungi tested was completely inhibited at 250 ppm. The good antifungal activity reported for the extracts of *P. amboinicus* can be attributed to the presence of two or more terpenic compounds [32]. For the case of the antifungal effect of *S. aromaticum* essential oil against *Fusarium* species, Sharma et al. [10] found that at a concentration of 250 ppm, the mycelial growth of *F. oxysporum* was completely inhibited. In this study, a doubling of the concentration was required to achieve full inhibition of the same pathogen; this difference might be due to variations in the chemical composition of the essential oils since previous studies have attributed variations to various factors, such as environmental conditions, geographic location, and harvest time, among others [33]. Since the *P. amboinicus* essential oil was the best at inhibiting *F. oxysporum*, its chemical composition was analyzed.

The chemical composition of the *P. amboinicus* essential oil was identified by using GC–MS, where the main compounds responsible for the antifungal activity were found. In this investigation, the major chemical compound was carvacrol, which coincides with other investigations [30,34]. These monoterpene hydrocarbons form an important part of the essential oil. These major constituents are non-polar compounds, which means that the hydrophobicity of the essential oil allows them to break through the lipids of the cell membrane and mitochondria, making them permeable and causing leakage of cell content, which leads to the inactivation of the pathogen [23]. The accumulation of some essential oil components in the lipophilic hydrocarbon molecules of the cell lipid bi-layer may also facilitate the transfer of other essential oil constituents into the cell [27]. There are some variations of these compounds reported in other studies [23,35]. It is known that the variation in chemical composition may be due to geographical and seasonal variation, the age of the plant, or the extraction method [36]. The chemical analysis of *P. amboinicus* essential oil was studied for its in vitro antibacterial activity against pathogenic bacteria contaminating food [37]. This research identified nine compounds, which were basically formed of monoterpenes and sesquiterpenes. The major compounds observed included thymol (45.64%), followed by p-cymene (19.46%), β-myrcene (12.59%), and α-terpinolene (9.86%). Another research identified caryophyllene, caryophyllene oxide, aromadendrene oxide, and selinene as the major chemical compounds in *P. amboinicus* essential oil. The researchers concluded that environmental conditions possibly caused changes in the chemical composition [23].

It is necessary to directly evaluate the effect on plants, such as ornamentals plants, using concentrations of 300 to 400 μL/L of the *P. amboinicus* essential oil; this is an excellent option since, with these doses, it is feasible to control various diseases or pests that affect the plants. Finally, it is possible to obtain good yields in less time than using conventional hydrodistillation; therefore, the use of microwaves represents, above all, a lower energy expenditure, contributing to taking care of the environment. All the essential oils present antifungal activity against *F. oxysporum*; however, the essential oils of *P. amboinicus* and *S. aromaticum* achieve 100% inhibition, which rate depends on the doses of the plant extracts.

## 5. Conclusions

The use of microwave technology to obtain plant extracts represents a more sustainable option to conventional methods. At the maximum concentration evaluated of EOs of *P. amboinicus* and *S. aromaticum*, there was a 100% inhibition of the pathogen under study. The chemical analysis of the essential oil of *P. amboinicus* revealed the presence of the compounds responsible for the biological activity and its potent antifungal effect. The EOs of *P. amboinicus* and *S. aromaticum* represent a safer and more effective option than polluting fungicides and are potential biological control agents against *F. oxysporum*.

## Figures and Tables

**Figure 1 life-13-00528-f001:**
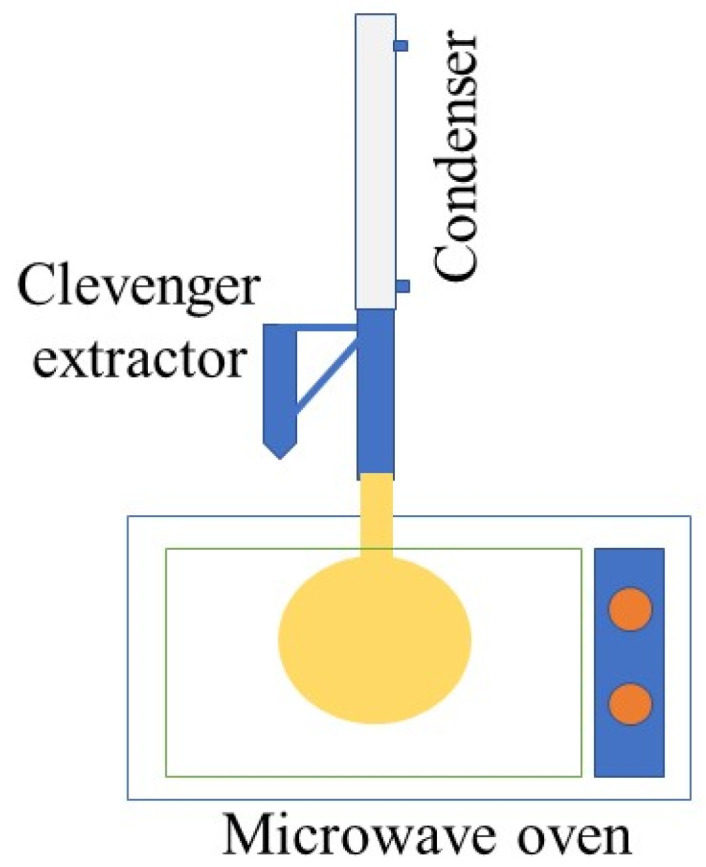
Microwave-assisted hydrodistillation (MAH) system adapted for the extraction of essential oils (EOs).

**Figure 2 life-13-00528-f002:**
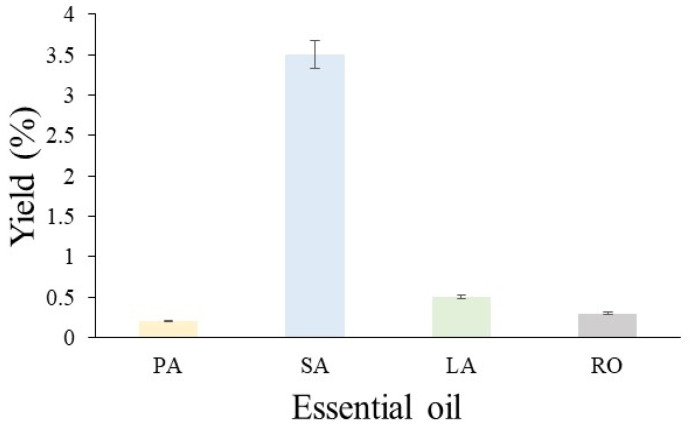
Essential oil yields from *P. amboinicus* (PA), *S. aromaticum* (SA), *L. alba* (LA), and *R. officinalis* (RO) obtained by using MAH.

**Figure 3 life-13-00528-f003:**
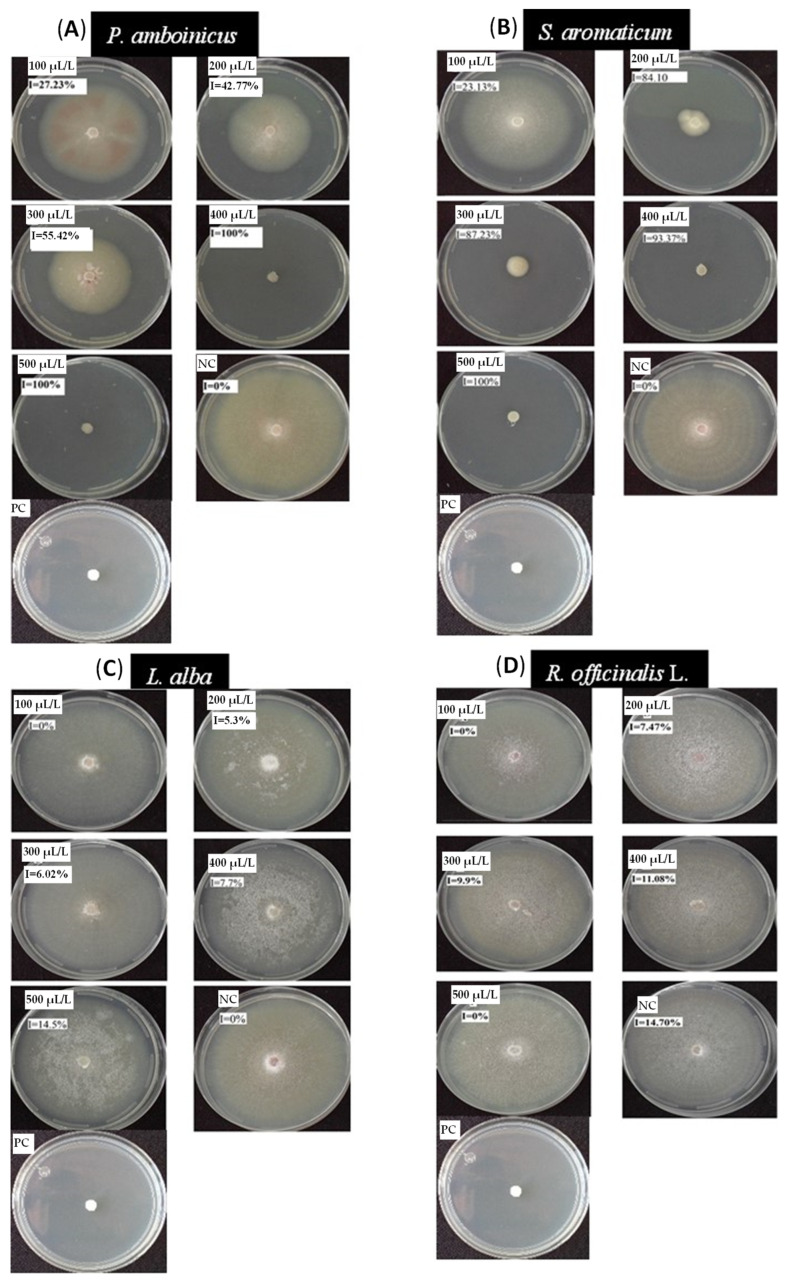
Effect of different doses of plant extracts on the mycelial growth of F. oxysporum at 10 days of incubation. (**A**) *Plectranthus amboinicus*, (**B**) *Syzygium aromaticum*, (**C**) *Lippia alba* and (**D**) *Rosmarinus officinalis*. I: % inhibition rate. PC: positive control (Benomyl 0.03 g/L). NC: negative control.

**Table 1 life-13-00528-t001:** Effect of essential oils at different doses on the inhibition of mycelial growth (%) of *F. oxysporum* after 10 days of incubation.

Inhibition (%) at Different Doses (µL/L)					
Essential Oil	100	200	300	400	500
*P. amboinicus*	27.2 ± 0.2 ^f^	42.4 ± 0.3 ^d^	55.7 ± 0.4 ^e^	100.0 ± 0.1 ^a^	100.1 ± 0.1 ^a^
*S. aromaticum*	23.1 ± 0.3 ^f^	84.1 ± 0.1 ^c^	87.3 ± 0.2 ^d^	93.3 ± 0.2 ^b^	100.0 ± 0.0 ^a^
*L. alba*	0.0 ± 0.1 ^i^	6.2 ± 0.1 ^hi^	5.3 ± 0.3 ^i^	7.7 ± 0.2 ^i^	14.5 ± 0.1 ^g^
*R. officinalis*	0.0 ± 0.0 ^i^	7.4 ± 0.0 ^h^	9.9 ± 0.1 ^h^	11.0 ± 0.1 ^h^	14.7 ± 0.3 ^g^

Values are expressed as the means of triplicate experiments ± standard deviation. Common letter means are not significantly different according to ANOVA (*p* < 0.05).

**Table 2 life-13-00528-t002:** Chemical composition of *Plectranthus amboinicus* essential oil, identified by GC–MS.

No	RT ^1^	Compound	% ^2^
1	3.489	(E)-beta-Famesene	0.45
2	3.774	beta-Myrcene	0.97
3	3.95	o-Cymene	11.61
4	4.15	gamma-Terpinene	6.91
5	4.325	5-Isopropyl-2-methylbicyclohexan-2-ol	0.43
6	4.795	Terpinen-4-ol	1.35
7	5.401	3-Methyl-2-(2-methyl-2-butenyl)-furan	1.26
8	5.581	Carvacrol	41.20
9	5.708	Ascaridole epoxide	0.19
10	6.045	Phenol, 2-methoxy-3-(2-propenyl)-	0.07
11	6.165	Cyclohexene, 2-ethenyl-1,3,3-trimethyl-	1.03
12	6.784	11,11-Dimethyl-4,8-dimethylenebicycloundecan-3-ol	0.21
13	7.032	Caryophyllene	11.45
14	7.102	α-Bergamotene	7.71
15	7.224	(1R,7S,E)-7-Isopropyl-4,10-dimethylenecyclodec-5-enol	0.38
16	7.361	1,4,7, -Cycloundecatriene, 1,5,9,9-tetramethyl-, Z, Z,Z-	3.76
17	7.499	(1R,7S, E)-7-Isopropyl-4,10-dimethylenecyclodec-5-enol-Dup1	1.03
18	7.611	(E)-beta-Famesene-Dup1	0.41
19	7.797	beta-Bisabolene	0.87
20	7.963	3,8,8-Trimethyl-6-methyleneoctahydro-1H-3a,7-methanoazulen	0.50
21	8.66	Caryophyllene oxide	4.62
22	8.915	(1R,3E,7E,11R)-1,5,5,8-Tetramethyl-12-oxabicyclo dodeca-3,7-diene	0.88
23	9.176	11,11-Dimethyl-4,8-dimethylenebicyclo [7.2.0] undecan-3-ol-Dup1	0.63
24	9.39	(1R,7S, E)-7-Isopropyl-4,10-dimethylenecyclodec-5-enol-Dup2	0.53
25	9.53	(1R,7S, E)-7-Isopropyl-4,10-dimethylenecyclodec-5-enol-Dup3	1.19
26	11.64	Naphthalene, 1,2,3,4,4a,5,6,7-octahydro-4a-methyl-	0.19
27	12.24	Naphthalene, 1,2,3,4,4a,5,6,7-octahydro-4a-methyl--Dup1	0.20

^1^ Retention time. ^2^ Percentage of total.

## Data Availability

The data presented in this study are available from the corresponding author upon request.

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
