# Peer review of "Microwave-Assisted Hydrodistillation of Essential Oil from Plectranthus amboinicus: Evaluation of Its Antifungal Effect and Chemical Composition"

_life, 2023, doi:10.3390/life13020528_

Round 1

Reviewer 1 Report

Dear Authors 

you should added a conclusion of your work in abstract 

some scientific name in the text should be italic 

when you write the scientific name of the plant only in the first time should be full name for the genus and then you can write the first litter only

my main question is how many replicate you used in the experiment 

and how you identified the fungi  

the Fig should be improve 

Reviewer 2 Report

The communication entitled "Microwave-assisted extraction of four essential oils: evaluation of its antifungal effect and chemical composition" is very interesting from the point of view of organic protection of plants from disease agents Fusarium oxysporum.

There are some comments.

Instead EOS - EOs

Plant material - who determined plant material, where it is deposited, voucher No? Is this wild growing plants or cultivated?

2.3. EOS extraction - provide more information! Is this some type of house-built apparatus? Give scheme!

Time of extraction? Same for all species?

What about Fusarium oxysporum? Is this your isolate? give more information.

Reviewer 3 Report

You should mention the plants in the title.

Abstract very poor written please add more highlights of your work.

Introduction  add part about importance of Medicinal Plants.

Results you should add positive control.

Discussion you should relate with other literature with gcmass.

Round 2

Reviewer 3 Report

Improvement in discussion section.

Please refer to previous works.
